# Harnessing the Sorghum Microbiome for Enhancing Crop Productivity and Food Security Towards Sustainable Agriculture in Smallholder Farming

**DOI:** 10.3390/plants14213242

**Published:** 2025-10-22

**Authors:** Omolola Aina, Lara Donaldson

**Affiliations:** 1International Centre for Genetic Engineering and Biotechnology (ICGEB), Cape Town 7925, South Africa; aina.temilola@icgeb.org; 2Department of Molecular and Cell Biology, University of Cape Town, Rondebosch 7700, South Africa

**Keywords:** microbiome, sorghum, smallholder farming, plant growth-promoting rhizobacteria, plant growth-promoting endophytes

## Abstract

Smallholder farming plays a crucial role in global food security, contributing more than half of the world’s food supply. However, it is increasingly threatened by climate variability, declining soil fertility, and financial constraints, all of which suppress plant growth, reduce yields, and endanger livelihood stability. Addressing these challenges requires sustainable, eco-friendly alternatives to costly and environmentally damaging agrochemicals. Sorghum, a climate-resilient cereal, harbours a diverse microbiome that contributes significantly to its remarkable adaptability under adverse conditions. Harnessing the sorghum-associated microbiome, therefore, represents a promising, low-cost, and sustainable strategy to enhance sorghum productivity and resilience in smallholder farming systems. However, despite its potential, the adoption of microbiome-based technologies among smallholders remains limited due to a lack of local production units, poor government policies, knowledge gaps, and perceived risks. Although many studies report positive outcomes from microbiome-based interventions, translating this potential from controlled experiments to real-world field applications requires a critical evaluation of the efficacy, practicality, and limitations of microbial interventions. Furthermore, the outcomes of these studies are uneven, highly context-dependent, and often restricted to short-term or small-scale trials. This review, therefore, seeks to highlight current understanding of the sorghum microbiome, including its composition and the procedures for isolating and characterising beneficial microbes. It further evaluates the key challenges hindering adoption and proposes strategies to overcome them. Ultimately, this review advocates for integrating sorghum-associated microbiome technologies within integrated farming systems, underscoring their potential to enhance sustainable crop production, strengthen smallholder resilience, and contribute to the global sustainable development goals.

## 1. Introduction

Smallholder or subsistence farming refers to a family-oriented agricultural system typically practised on a small area of land, generally ranging from less than one hectare to 10 hectares. These systems rely predominantly on household labour and are characterised by low external input, with a portion of the agricultural output reserved for subsistence needs [1]. Despite their modest scale, smallholder farms account for approximately 90% of the world’s farmland use. They are responsible for producing nearly one-third of the global food supply, underscoring their critical role in ensuring global food security and sustaining rural livelihoods [2].

Smallholder farming systems make significant contributions to socioeconomic development in developed and developing countries by enhancing job security, alleviating poverty, promoting crop diversification, and fostering self-sufficiency [3]. In sub-Saharan Africa, they account for approximately 90% of food production, thereby helping to lower food prices in both rural and urban areas. Moreover, these farming systems play a vital role in national economies, contributing approximately 70% to GDP by increasing household incomes and generating over 50% of employment opportunities [4].

Smallholder farming systems are major stakeholders in ensuring the world’s predicted population of nine billion is fed by 2050 [5]. However, these farmers face several challenges, including climate change, soil infertility, pest and disease outbreaks, water scarcity, and soil degradation, all of which reduce crop yields [6]. These issues are traditionally addressed through the use of chemical fertilisers, pesticides, and irrigation to enhance productivity. While effective in the short term, these methods are often expensive and inaccessible to resource-limited smallholder farmers. Additionally, prolonged dependence on synthetic inputs is environmentally unsustainable and can degrade soil health, reduce biodiversity, and further exacerbate productivity challenges [7].

Given the significance of smallholder farming in ensuring food security and socioeconomic growth, there is an urgent need to adopt sustainable agricultural practices that enhance crop productivity while preserving soil health and maintaining ecosystem balance [8]. One promising approach involves leveraging the plant microbiome to improve crop growth and resilience [9].

In recent decades, research has underscored the vital and complex role of the plant microbiome in promoting plant growth, health, and resilience. The plant microbiome comprises both soil microbial communities, the endosphere, and the phyllosphere [10]. Plants regulate the composition of their microbiome through various signalling mechanisms and interactions with soil chemistry. The root exudates and soil biochemistry create a specialised environment that facilitates the recruitment of plant-specific rhizosphere communities based on prevailing environmental conditions [11]. From this diverse microbial pool, plants selectively form symbiotic or endophytic relationships with microbes that possess beneficial biochemical traits [12]. Among the most well-studied symbionts are plant-growth-promoting rhizobacteria (PGPR), along with mycorrhizal fungi, including ectomycorrhizal (EM) and arbuscular mycorrhizal (AM) fungi [11].

Sorghum (*Sorghum bicolor* Moench) is one of the world’s most important cereal crops, ranking among the top five cultivated globally, with a land area of 42.3 million hectares and a yield of 61.5 million metric tons [13]. It serves as a main staple diet for over 300 million people and a source of income for millions of smallholder farmers, especially in developing countries such as sub-Saharan Africa [14]. Sorghum has been reported to possess some characteristics which make it well-adapted and able to thrive in harsh climatic conditions. These include a waxy bloom on the leaves that reduces water loss, an extensive root system, C4 photosynthesis, the ability to stop growth under stress and resume it when the stress is relieved, and a diverse microbiome community (Zheng et al., 2024 [15]). Its remarkable ability to withstand harsh environmental conditions makes it an ideal candidate for harnessing plant-associated microbes to improve crop resilience and productivity [16].

Recent research has positioned sorghum as a model crop for understanding drought tolerance mechanisms, highlighting its interactions with beneficial microbiomes [17]. Microbes inhabiting different compartments of sorghum, including roots, leaves, and the rhizosphere, play a vital role in enhancing stress tolerance and improving yield [18]. For example, Liu and colleagues identified fungal and bacterial communities, such as *Confluentibacter*, *Lysobacter*, and *Chaetomium*, in different compartments of sorghum plants during vegetative and reproductive stages. Their findings highlighted that the enhanced diversity and stability of these microbial communities play a significant role in improving sorghum grain yield and protein content by facilitating increased nitrogen fixation and phosphorus solubilisation [19].

Despite its promising potential, leveraging the sorghum microbiome for sustainable agriculture faces several challenges. The complexity of microbial interactions and the dynamic nature of environmental conditions limit the effectiveness and persistence of beneficial microbes in field settings [20]. Furthermore, gaps in our understanding of microbiome functionality and technological constraints in isolating and maintaining key microbial strains have hindered the development of practical microbiome-based solutions for sorghum improvement [20].

Thus, in this review, we discuss the significance of the sorghum microbiome in enhancing sorghum resilience and productivity. We highlight the knowledge gaps and priority areas in sorghum microbiome research to improve agricultural outcomes in smallholder farming systems. By deepening our understanding of sorghum-microbe interactions, we aim to highlight innovative microbiome-driven strategies that can enhance food security and promote climate-resilient agricultural systems, particularly smallholder farming systems. In addition, this review also expands the knowledge base of smallholder farmers, plant biologists, and policymakers on the potential of the sorghum-associated microbiome to maximise crop yield towards sustainable food production for the ever-growing human population.

## 2. Methodology

This review was conducted using a structured literature search to ensure transparency and reproducibility. Articles were retrieved from Web of Science, Scopus, and Google Scholar databases covering the period 2000–2025. Search terms included combinations of “sorghum microbiome,” “plant-growth-promoting rhizobacteria (PGPR),” “plant-growth-promoting-endophytes,” “arbuscular mycorrhizal fungi,” “biofertilisers,” “microbial inoculants,” and “sustainable agriculture.” The search was initially focused on articles published between 2000 and 2025 to capture modern research, but seminal earlier works were also included. The retrieved articles were screened based on their titles and abstracts, and those selected for full-text review were assessed for their relevance to the core themes of the manuscript: microbial isolation, mechanistic insights, field efficacy, and practical application challenges. Priority was given to peer-reviewed original research articles, meta-analyses, and authoritative reviews. This process aimed to provide a comprehensive, critical, and balanced synthesis of the current state of knowledge, identifying key research gaps and future directions for harnessing the sorghum microbiome in sustainable agriculture.

## 3. Overview of the Sorghum Microbiome Community

The sorghum microbiome comprises a diverse array of microorganisms, including bacteria, archaea, fungi, oomycetes, viruses, nematodes, and protists [21]. These microbes interact intricately with the host plant in complex ways, exerting both detrimental and beneficial effects on plant health and development. Pathogenic microbes negatively influence plant physiology through mechanisms such as disrupting photosynthesis, manipulating hormonal signalling pathways, and inducing oxidative stress, often culminating in cell death. In contrast, beneficial microbes significantly contribute to plant growth and resilience by enhancing nutrient acquisition, improving tolerance to abiotic stressors, and providing defence against plant pathogens [22]. This review focuses specifically on the beneficial components of the sorghum microbiome.

The beneficial microbiome of sorghum colonises various plant tissues and can be broadly classified into distinct communities based on their spatial localisation, including the rhizosphere, endosphere, and phyllosphere, as depicted in Figure 1 [23]. The rhizosphere is a narrow zone of soil surrounding the root (typically within 1–10 mm), characterised by complex biochemical interactions between root exudates, soil particles, and microbial populations. This region is highly active and complex, hosting a diverse array of beneficial microorganisms [24]. The endosphere comprises the internal tissues of the plant, including roots, stems, leaves, seeds, and flowers. Beneficial organisms that inhabit this compartment, known as endophytes, establish symbiotic relationships with the host, contributing to stress mitigation, nutrient cycling, and enhanced productivity. In contrast, the phyllosphere encompasses all the aerial parts or above-surface organs of the sorghum plant (leaves, stems, flowers, and seeds), representing a highly dynamic and variable environment. Microbial communities here are exposed to fluctuating environmental conditions, including light, humidity, and temperature, which influence their composition and function [25].

Functionally, the beneficial sorghum microbiome can be categorised into major groups, such as plant growth-promoting rhizobacteria (PGPR), arbuscular mycorrhizal fungi (AMF), plant growth-promoting fungi (PGPF), and plant growth-promoting endophytes (PGPE). These groups contribute to sorghum health and productivity through nutrient mobilisation, phytohormone production, pathogen suppression, and enhanced stress resilience (Table 1).

Plant growth-promoting rhizobacteria (PGPR) are beneficial microbes commonly found in the rhizosphere, where they colonise plant roots and enhance plant development. These bacteria may exist freely in the soil or form symbiotic associations with plants [33]. PGPRs can be extracellular, residing in the rhizosphere or root cortex, or intracellular, colonising root cells. The common PGPR taxa associated with sorghum include members of the *Proteobacteria* (*Pseudomonas*, *Rhizobium*, *Azospirillum*, *Enterobacter*) and *Firmicutes* (*Bacillus*, *Paenibacillus*) [26]. These PGPRs enhance plant growth through various mechanisms, including (i) facilitating nutrient uptake, (ii) phosphate solubilisation, (iii) nitrogen fixation, (iv) synthesis of 1-aminocyclopropane-1-carboxylic acid (ACC) deaminase to reduce ethylene stress, (v) production of phytohormones (auxins, cytokinins, and gibberellins), and (vi) biocontrol activity against pathogens. They also enhance stress resilience in sorghum plants by stimulating the expression of antioxidant enzymes (superoxide dismutase, catalase, and ascorbate peroxidase), increasing the accumulation of plant osmoprotectants (e.g., proline, glycine betaine), and promoting the production of volatile organic compounds [27].

Arbuscular mycorrhizal fungi (AMF) are a group of symbiotic fungi that establish symbiotic relationships with the roots of sorghum plants. These fungi, primarily from the phylum *Glomeromycota*, colonise the root cortex and form specialised structures called arbuscules within cortical cells [26]. Arbuscules increase the surface area of contact between fungal hyphae and plant cell membranes, facilitating the efficient transfer of essential minerals, particularly phosphorus, potassium, zinc, iron, water, and other micronutrients that are often poorly available in the soil. In return, the plant supplies the fungi with carbohydrates derived from photosynthesis. This association significantly improves plant growth, particularly under abiotic stress conditions, by boosting nutrient efficiency and stress resilience [30].

Plant growth-promoting fungi (PGPF) associated with the rhizosphere and endosphere of sorghum play vital roles in enhancing crop performance and resilience. They establish beneficial interactions with sorghum roots, utilising root exudates to colonise the rhizosphere or internal tissues and contribute to sustainable growth through multiple functional traits, including the synthesis of phytohormones, carbon sequestration, enhanced nutrient uptake, disease suppression, and boosting the plant immune system [34]. Several fungal genera, including *Trichoderma*, *Fusarium*, *Aspergillus*, *Penicillium*, and *Talaromyces*, have been isolated from sorghum roots and the rhizospheric soils surrounding them. These fungi enhance nutrient availability by solubilising phosphorus and other minerals and promoting nitrogen cycling. For instance, *Trichoderma* spp. associated with sorghum have been shown to produce phytohormones, such as indole-3-acetic acid (IAA), and enzymes like cellulases and chitinases, which improve root architecture and suppress pathogen infection [35]. Moreover, these fungi can induce systemic resistance in sorghum, thereby enhancing tolerance to biotic stresses such as *Striga* infestation and fungal pathogens, as well as abiotic stresses like drought and salinity [36].

Plant growth-promoting endophytes (PGPE) are bacteria and fungi that inhabit the internal tissues of sorghum plants without eliciting pathogenic symptoms. These microbes establish close and often mutualistic relationships with their host, colonising various tissues, including roots, stems, leaves, and seeds. Unlike many rhizospheric organisms, endophytes reside within plant tissues, enabling direct interaction with host cells and facilitating efficient nutrient exchange [27]. This intimate association often leads to enhanced plant growth and resilience. Endophytes contribute to various plant physiological functions, such as improving nutrient uptake, synthesising phytohormones and siderophores, and conferring protection against abiotic and biotic stresses. Bacterial endophytes commonly belong to the phyla *Proteobacteria*, *Firmicutes*, and *Actinobacteria*, with representative genera including *Bacillus*, *Pseudomonas*, *Burkholderia*, *Micrococcus*, *Stenotrophomonas*, and *Pantoea*. Endophytic fungi, on the other hand, are frequently members of the *Basidiomycota* and *Ascomycota*, forming associations that support host plant growth and stress tolerance [26,37].

Various mechanisms have been suggested as ways by which sorghum selectively recruits beneficial microbes. These involve the use of plant exudates, including sugars, flavonoids, organic acids, growth factors, vitamins, strigolactones and other chemoattractants that selectively attract microbes with beneficial traits, such as nitrogen-fixing bacteria and mycorrhizal fungi, while simultaneously deterring pathogens [38,39]. Sorghum plants also employ pattern recognition receptors (PRRs) on their cell surfaces to detect microbial-associated molecular patterns (MAMPs). This detection enables differential immune signalling that permits symbiosis with beneficial microbes while triggering stronger, defence-related responses against pathogenic microbes [40]. Additionally, root mucilage and border cells create a protective niche that supports the colonisation of beneficial microbes [41].

## 4. Prospects of the Sorghum Microbiome for Improved Crop Productivity in Smallholder Farming

Smallholder farming systems face several constraints that severely limit productivity and sustainability. Climate variability, characterised by rising temperatures, erratic rainfall, and recurrent droughts, increasingly disrupts critical crop developmental stages [42,43]. Financial barriers compound these challenges, as limited access to credit, fluctuating market prices, and inadequate governmental support restrict farmers’ ability to invest in essential inputs, including improved sorghum varieties, fertilisers, and pesticides [44,45]. Furthermore, short-term production goals and immediate livelihood needs often drive unsustainable agronomic practices, leading to long-term declines in soil fertility and crop productivity [46].

Previous research indicates that the sorghum-associated microbiome, particularly PGPR, AMF, PGPF, and PGPE, holds considerable promise for alleviating key constraints in smallholder farming systems and enhancing sorghum productivity. For example, Mareque et al. [47] demonstrated that several PGPE strains (*Rhizobium* spp., *Pantoea* spp., *Enterobacter* spp., and *Bacillus* spp.), isolated from sorghum roots and stems, increased shoot, root, and stem biomass by 10–35% under a four-month greenhouse experiment. Likewise, a greenhouse pot study by Da Silva et al. [48] reported that inoculating sorghum plants with PGPE (*Enterobacter* spp., *Klebsiella* spp., and *Pantoea* spp.) sourced from field-grown sorghum in Brazil enhanced shoot biomass, elevated nitrogen concentration, and increased nitrogen accumulation at 51 days after emergence. In a separate study, native *Trichoderma* spp. isolated from rhizospheric soils across Uttarakhand, India, displayed significant biocontrol potential against *Colletotrichum graminicola*: inoculation reduced disease severity by approximately 54% and increased grain yield by up to 27% over two consecutive seasons [49].

A variety of well-characterised mechanisms mediates the observed beneficial effects. The most consistently documented mechanisms across studies in sorghum for growth promotion include phytohormone production (e.g., IAA and gibberellic acid (GA) biosynthesis), siderophore production, and improved nutrient acquisition via nitrogen fixation and phosphate solubilisation [50,51,52]. For biotic stress resistance, beneficial microbes often confer protection through direct antagonism (mycoparasitism), induction of systemic resistance in the host plant, and activation of defence pathways such as lignification and antioxidant enzyme production [31,53]. Abiotic stress tolerance, conversely, is frequently enhanced through synergistic pathways, including improved morphological processes (e.g., total chlorophyll content), biochemical signalling (e.g., ACC deaminase activity to lower stress ethylene), and the accumulation of osmolytes and antioxidants [54,55]. For smallholder farmers, mechanisms that directly improve nutrient availability and water-use efficiency may offer the most immediate and tangible benefits, as they address the most fundamental constraints in low-input systems. Table 2 summarises additional sorghum-microbe interaction studies, highlighting the microbial taxa examined, their proposed modes of action, and the resulting improvements in plant growth or stress tolerance.

Nevertheless, it is worth noting that many of these sorghum-associated microbial inoculants studies have been conducted under controlled greenhouse or in vitro conditions, with relatively few validated in multi-season field trials. While these microbes consistently enhanced biomass accumulation, nutrient acquisition, and stress tolerance, the majority of experiments were short-term, site-specific, and lacked geographic replication, resulting in an insufficient understanding of long-term host–microbe–environment interactions. This highlights a significant gap between experimental promise and practical application. The critical challenges of economic feasibility, scalable formulation, practical application methods, ecological risk assessment, and farmer-centric adoption pathways, which are required for their successful implementation in smallholder systems, are addressed in subsequent sections.

## 5. Procedures and Guidelines to Facilitate Successful Application of Sorghum Microbiome in Smallholder Farming

Effective application of sorghum-associated microbial inoculants in smallholder farming systems requires a systematic framework that ensures both microbial efficacy and farmer accessibility. The following procedures and guidelines are proposed to facilitate the successful integration of sorghum microbiome-based technologies in resource-limited settings:

### 5.1. Isolation, Characterisation and Efficacy Testing of Native Microbial Strains Under Controlled and Field Conditions

The successful utilisation of microbial inoculants in sorghum cultivation begins with the systematic isolation and characterisation of native microbial strains. This process involves targeted sampling of rhizospheric and endophytic microbial communities from healthy sorghum plants across diverse agroecological zones. These site-specific collections are crucial for capturing microbial diversity adapted to local soil conditions and climatic stresses [62]. Following isolation, microbial strains are screened for key plant growth-promoting (PGP) traits, including nitrogen fixation, phosphate solubilisation, ACC deaminase activity, IAA production, siderophore secretion, and biocontrol activity against known pathogens. These functional traits are crucial for enhancing nutrient availability, improving tolerance to abiotic stress, and suppressing disease in sorghum [63]. To ensure precision and strain-specific application, molecular characterisation should be performed using 16S rRNA gene sequencing (for bacteria) and ITS region sequencing (for fungi). This step facilitates the identification of dominant and potentially beneficial taxa. Once promising strains are identified, their efficacy must be rigorously evaluated through a two-tiered testing approach [64]. Greenhouse trials provide controlled environments to assess the influence of inoculants on sorghum growth, nutrient uptake, and resilience under abiotic and biotic stressors. Subsequently, field trials are conducted across multiple smallholder plots to validate these strains under real-world conditions. These trials are essential for determining the consistency of microbial performance, their interaction with existing agronomic practices, and their overall contribution to sorghum yield enhancement [65].

### 5.2. Formulation and Application Strategies for Sorghum Microbial Inoculants

The effectiveness of microbial inoculants in agricultural systems largely depends on their ability to survive and colonise suitable soil niches. Formulation is a key determinant of this success, as it ensures microbial survival during storage, transportation, and field application. It also enhances safety, handling, and compatibility with farm machinery. Poor formulation can result in the loss of up to 90% of applied microbes after application [66]. For successful adoption in smallholder sorghum systems, formulation and delivery strategies must emphasise cost efficiency, ease of use, and sustained microbial viability. This process begins with selecting appropriate carriers, preferably those that are low-cost and locally available. Carriers such as compost, vermiculite, biochar, molasses, and rice bran have proven effective in maintaining microbial stability and enhancing field applicability. These substrates support microbial survival during storage and transport, thereby improving the adaptability of inoculants to soil environments upon application and their interaction with native microbiomes [65].

For microbial inoculants to be a viable alternative, they must be formulated as stable products that can withstand extended storage. Depending on the formulation and storage environment, shelf-life expectations can range from two to three months at ambient temperature to more than a year under optimal conditions [67]. For carriers, a good moisture-absorbing capacity with a near-neutral pH is optimal for most bacteria [68]. Maximising the initial concentration of viable cells in the product is a common strategy to offset the inevitable decline in cell numbers during storage [69]. Nevertheless, storage parameters, including temperature, moisture, and aeration, should be carefully optimised to support long-term cell viability. Regulatory quality standards, which vary slightly across countries, typically specify microbial loads ranging from 10^7^ to 10^9^ colony-forming units (CFU) per gram or millilitre of formulation. Alternatively, some standards focus on viable cell numbers per seed after application, with recommended minimum levels typically around 10^4^–10^5^ CFU per seed [70].

From an economic standpoint, the use of locally available carriers, such as compost, rice bran, or molasses, helps keep bioinoculant production costs low, offering a clear advantage for smallholder farmers [71]. Evidence from West Africa shows that the biofertiliser NoduMax costs approximately US$5 per hectare, compared to nearly US$100 per hectare for urea fertiliser, which is needed to provide an equivalent nitrogen supply [72]. Similar trends have been reported in Zimbabwe and Rwanda, where peat-based *Rhizobium* inoculants sufficient for one hectare were priced between US$4.50 and 5.20, substantially lower than the cost of mineral fertilisers delivering the same nutrient levels [73]. Collectively, these findings underscore the potential of microbial inoculants to significantly reduce input costs and enhance affordability in resource-constrained farming systems.

Equally crucial to formulation is the application strategy, which must ensure precise delivery of the inoculant at the right time, in the right quantity, and at the appropriate site [74]. Beneficial microbes are most effective when introduced early in the plant’s life cycle, particularly during germination and root growth stages. Early-stage inoculation enhances root colonisation, facilitates nutrient acquisition, and improves resilience to both biotic and abiotic stresses [75].

To maximise the effectiveness of microbial inoculants, it is essential to carefully design application methods that optimise interactions between plants and beneficial microbes [74]. Among the various techniques, seed biopriming applied as a slurry or through pre-coated treatments, is widely recognised for its simplicity, cost-effectiveness, and consistent performance. This method promotes early microbial colonisation with minimal quantities of inoculant, making it especially suitable for resource-constrained farming systems [76]. Another effective strategy is soil drenching, where the inoculant is applied directly to the root zone either at planting or during the early vegetative stage. This approach ensures the rapid establishment of microbial communities in the rhizosphere, thereby supporting plant growth and resilience from the outset [77].

### 5.3. Monitoring, Evaluation, and Regulatory Compliance for Sustainable Use of Sorghum Microbiome-Based Inoculants

Implementing sorghum microbiome technologies in smallholder systems requires robust monitoring, evaluation, and regulatory frameworks to ensure efficacy, farmer satisfaction, and environmental safety [78]. Simple and farmer-friendly monitoring tools should be developed to effectively monitor the performance of microbial inoculants across diverse agroecological contexts. Digital tools that can be easily accessed by smallholder farmers, such as smartphone applications, should be used to replace traditional paper-based methods. These digital tools can facilitate the systematic documentation of changes in crop growth, yield, stress tolerance, and overall farmer experience, thereby enabling more accurate, timely, and scalable evaluation of inoculant effectiveness under field conditions [79]. Establishing robust feedback mechanisms is essential for continuously improving microbial inoculant technologies. The data obtained from field monitoring should be systematically analysed to inform the refinement of microbial strains, enhance formulation protocols, optimise application strategies, and adapt extension messaging to local contexts [80]. Actively involving farmers in this feedback loop enhances the relevance of microbiome interventions, builds trust, promotes a sense of ownership, and ultimately supports the long-term adoption and sustainability of these technologies [81].

Equally important is adherence to regulatory standards. All microbial strains and formulations must comply with national biosafety and agricultural input regulations to ensure they are safe for the environment, non-pathogenic to humans and animals, and ecologically compatible with existing soil microbiota [74]. In this regard, the Southern African Development Community (SADC) has played a key role by encouraging harmonised regulatory frameworks across member states for biopesticides and microbial inoculants. By aligning national biosafety and agricultural input regulations, SADC fosters consistent standards for microbial products, ensuring they meet safety, environmental, and efficacy criteria. These regional efforts are crucial in supporting the development of quality assurance protocols that ensure the integrity of microbial formulations, building trust among farmers and promoting the use of biological alternatives in agriculture [82]. Furthermore, establishing quality assurance protocols, including routine testing of inoculant viability, microbial purity, and contaminant levels, is critical before distribution. These standards safeguard product integrity and maintain farmer confidence in using biological alternatives to chemical inputs [83].

### 5.4. Integration with Sustainable Farming Practices

Maximising the potential of sorghum microbiome-based inoculants requires their integration into the broader sustainable agricultural systems [65]. The synergistic use of microbial inoculants alongside existing low-input practices commonly employed by smallholder farmers, such as the use of organic soil amendments (e.g., compost, green manure, and farmyard manure), can significantly enhance microbial survival, improve nutrient availability, and increase soil organic matter content [84]. Furthermore, combining inoculants with conservation tillage, crop rotation, and intercropping promotes greater biodiversity and resilience within the agroecosystem. This integrated approach aligns with the United Nations Sustainable Development Goals (SDGs) by advancing environmental sustainability, food safety, and climate resilience while ensuring stable and productive agricultural outputs [85].

For instance, reduced tillage preserves microbial habitats and minimises soil disturbance, while intercropping sorghum with legumes or compatible cover crops can improve nitrogen fixation and suppress pests and diseases. Rotational schemes that alternate host and non-host crops can also reduce pathogen pressure and enhance soil fertility [86,87].

Smallholder farmers can build more productive, adaptive, and environmentally sustainable farming systems by embedding microbial inoculant use within a holistic agroecological framework. This approach amplifies the agronomic effects of microbiome interventions, contributing to the broader global food security goals and ecological sustainability under climate variability [88].

## 6. Isolation and Characterisation of the Sorghum Microbiome

Effective utilisation of the sorghum microbiome requires an integrated pipeline that combines conventional culture-based or traditional isolation techniques with modern molecular and omics-based tools. Such a multi-tiered approach is essential for identifying, prioritising, and developing microbial strains into field-ready products for smallholder farming systems [89]. Conventional culture-dependent isolation typically begins with sampling different plant compartments, including rhizosphere soil adhering to roots, phyllosphere surfaces (such as leaves, stems, and seeds), and internal plant tissues, to access microbial diversity [90]. Standard practices, such as selective or general-purpose media, serial dilution, and streak or spread plating, remain widely used to recover morphologically distinct colonies, which are then purified by subculturing and preserved for downstream analyses [91,92]. These approaches have been successfully applied to sorghum-associated microbes with plant-growth-promoting or biocontrol properties; for example, Martinez et al. [93] recovered antifungal bacteria from the sorghum rhizosphere, Chandra et al. [94] isolated arbuscular mycorrhizal fungi with salinity-alleviating potential, and 48 endophytic strains with growth-promoting traits were obtained from sweet sorghum tissues [95].

However, conventional culture-dependent isolation only captures a fraction (less than 1%) of the total microbial diversity present in sorghum-associated habitats [96]. It also overlooks slow-growing or syntrophy-dependent organisms that cannot be cultured under standard in vitro conditions [97]. As a result, the structural and functional complexity of the sorghum microbiome, along with the associated beneficial traits, remains underestimated [98]. Nonetheless, culture-based isolation remains indispensable for obtaining live strains for functional testing and large-scale production. It also enables phenotypic assays, safety evaluation, and scalable production of candidate inoculants [99].

To overcome the inherent limitations of conventional culture-dependent techniques, molecular and culture-independent approaches are now routinely employed to profile both the culturable and unculturable fractions of the sorghum microbiome [99]. These include molecular fingerprinting methods such as denaturing gradient gel electrophoresis (DGGE) and terminal restriction fragment length polymorphism (T-RFLP), as well as high-throughput sequencing [100]. Standard workflows typically involve total DNA extraction from plant or rhizosphere samples, polymerase chain reaction amplification of conserved taxonomic markers (e.g., bacterial 16S rRNA, fungal 18S rRNA, or ITS regions), and subsequent sequence comparison against curated databases such as GenBank, SILVA, or UNITE [92]. For example, a 16S rRNA PCR–RFLP approach was used to characterise 280 bacterial isolates with plant-growth-promoting traits from sorghum roots [101]. Similarly, DGGE, T-RFLP and molecular sequencing have been applied to detect and differentiate *nifH*-harbouring bacterial taxa in the sorghum rhizosphere, facilitating the targeted recovery of putative nitrogen-fixing strains [102]. In parallel, sequencing of the 18S and ITS regions has revealed diverse assemblages of arbuscular mycorrhizal fungi associated with sorghum roots, which are implicated in enhanced tolerance to abiotic and biotic stresses [94,103].

Recent advances in omics technologies, including metagenomics, transcriptomics, proteomics, lipidomics, and metabolomics, are now reshaping sorghum microbiome research [104]. They enable high-resolution, functional profiling of entire microbial communities under different environmental conditions [105]. Building on these approaches, Hara et al. integrated metagenomic and proteomic analyses to identify nitrogenase genes and proteins and successfully recovered functional nitrogen-fixing bacteria associated with four field-grown sorghum genotypes [106]. In parallel, complementary chemical analyses, such as Gas Chromatography-Mass Spectrometry (GC-MS), Liquid Chromatography-Mass Spectrometry (LC-MS), and Nuclear Magnetic Resonance (NMR), further reveal the bioactive metabolites (e.g., siderophores, phytohormones, and volatile organic compounds) secreted by these microbes, which contribute to plant health and resilience [99,107]. For instance, Wang, et al. [108] employed GC-FID (Gas Chromatography with Flame Ionisation Detection) and metabolomic analysis to identify shifts in the sorghum rhizosphere microbiome, which were influenced by changes in the secretion of root exudates, such as sorgoleone. Similarly, GC-MS and metabolomics were used to characterise the microbial taxa associated with the sorghum rhizosphere in response to intercropping with peanut under salt stress conditions [109]. Taken together, integrating culture-dependent isolation with omics-based characterisation offers a robust framework for moving beyond descriptive studies towards targeted discovery and functional understanding that will better enable development of robust sorghum bioinoculants tailored to smallholder systems.

## 7. Challenges and Future Directions

Despite the promising potential of sorghum microbiome-based inoculants, several factors continue to constrain their successful adoption in smallholder farming systems. A significant limitation is the lack of local production units. Smallholder farmers face significant challenges accessing viable and affordable inoculants without decentralised, community-based production facilities [88]. Furthermore, poor storage conditions and limited shelf life of products exacerbate these challenges. Microbial inoculants are often sensitive to environmental stressors, including temperature fluctuations, desiccation, and ultraviolet (UV) radiation [83]. In rural areas, where cold-chain infrastructure is typically unavailable, inadequate storage facilities frequently result in a rapid decline in microbial viability before the products can be applied in the field [70,110].

Another critical consideration is the ecological implications of introducing microbial inoculants into the soil ecosystem. The application of microbial inoculants can alter the microbial community structure and function through competitive exclusion, where the inoculated strains outcompete native microbes for limited resources such as root exudates and nutrients [111]. Furthermore, the applied microbes may disrupt the established ecological equilibrium and modify the physicochemical environment through their metabolic activity, creating new niches that favour specific taxa over others. These shifts can have cascading effects, potentially reducing the diversity and functional redundancy of the indigenous community, which are key indicators of a resilient soil ecosystem [112]. Consequently, repeated application may lead to the suppression or displacement of beneficial native strains, inadvertently diminishing the very ecosystem benefits, such as natural pathogen suppression or nutrient cycling, that the inoculants are intended to enhance [113].

Furthermore, farmer reluctance remains a critical barrier to adoption. Due to knowledge gaps, perceived risks, and socioeconomic constraints, many smallholder farmers hesitate to adopt microbiome-based technologies to enhance sorghum productivity [114]. Microbial inoculants are often viewed with scepticism, particularly in comparison to conventional practices that offer more immediate and visible results. Moreover, smallholders generally operate under tight financial constraints and prioritise inputs with predictable and rapid returns. Since the benefits of microbiome inoculants, such as improved soil health, enhanced stress tolerance, and increased yield, may not be immediately observable, their use is often perceived as financially risky [115].

Government policies also play a role in influencing the uptake of microbiome technologies. Poorly designed or absent policies can create systemic barriers that hinder the development and use of sorghum microbiome-based solutions among smallholders [65]. In many regions, the lack of specific regulations recognising and supporting microbial inoculants leads to market uncertainty. Ambiguous registration processes, weak quality control standards, and the absence of biosafety guidelines discourage private-sector investment and erode farmer confidence [116]. Furthermore, smallholder farmers may struggle to invest in microbiome technologies without targeted subsidies or financial incentives, particularly when these must compete with subsidised chemical fertilisers and pesticides. Weak policy support for agricultural research and innovation also limits the development of locally adapted and effective sorghum microbiome strains [115].

Addressing these multifaceted challenges will require coordinated investment in localised, low-cost production systems and formulation technologies capable of extending shelf life without dependence on sophisticated storage infrastructure [74]. Long-term environmental monitoring using high-resolution tools such as amplicon sequencing and metagenomics is essential to ensure that the benefits of inoculation do not come at the cost of soil microbial biodiversity and long-term ecosystem stability. Emphasis should be placed on developing improved carrier materials, enhancing packaging technologies, and training farmers in proper inoculant handling to maintain product quality throughout the production and application process [83]. Additionally, strengthening local production and storage capacities is crucial for unlocking the full potential of sorghum microbiome inoculants in smallholder systems.

Moreover, participatory extension approaches, farmer-led trials, and community-based education programs must be prioritised to build trust, demonstrate efficacy, and align microbiome technologies with farmers’ existing practices and priorities [65]. Strengthening farmers’ scientific literacy regarding soil microbiomes and ensuring consistent access to high-quality, reliable inoculants will be pivotal to fostering adoption and maximising the impact of microbiome innovations [115].

Finally, strengthening policy frameworks to support microbiome research, ensure quality assurance, enhance financial accessibility, and promote farmer education is critical. [82]. Such efforts will facilitate the broader adoption of sorghum microbiome inoculants and contribute to more resilient, sustainable smallholder farming systems that align with long-term agricultural development goals [116].

## 8. Conclusions

The sorghum microbiome offers a promising and sustainable solution to the constraints limiting smallholder farming systems, including soil infertility, climate variability, pest and pathogen attacks, and limited access to inputs. By leveraging the natural symbioses between sorghum and its associated microbial communities, particularly plant growth-promoting rhizobacteria (PGPR), arbuscular mycorrhizal fungi (AMF), plant growth-promoting fungi (PGPF), and beneficial endophytes, it is possible to enhance nutrient acquisition, improve resilience to environmental stress, suppress diseases, and sustainably increase yields.

The development and application of native sorghum microbiome-based inoculants tailored to local agroecological conditions presents a cost-effective and environmentally friendly strategy that aligns with the goals of sustainable agriculture. However, successfully implementing these innovations requires addressing several key challenges, including establishing localised production units, improving inoculant formulation and storage stability, enhancing farmer awareness and training, and strengthening regulatory frameworks to support quality assurance and market access.

Integrating sorghum microbiome technologies into broader sustainable farming practices, such as organic amendments, conservation tillage, and diversified cropping systems, will further amplify their benefits and contribute significantly toward achieving global food security and the United Nations Sustainable Development Goals (SDGs). Future research should prioritise participatory approaches that involve farmers directly in technology development, ensure the adaptability of microbial inoculants under field conditions, and promote policy environments that encourage investment, education, and innovation.

Overall, the strategic harnessing of the sorghum microbiome represents a transformative pathway for empowering smallholder farmers, enhancing agricultural resilience, and promoting the transition toward more sustainable, productive, and climate-resilient farming systems.

## Figures and Tables

**Figure 1 plants-14-03242-f001:**
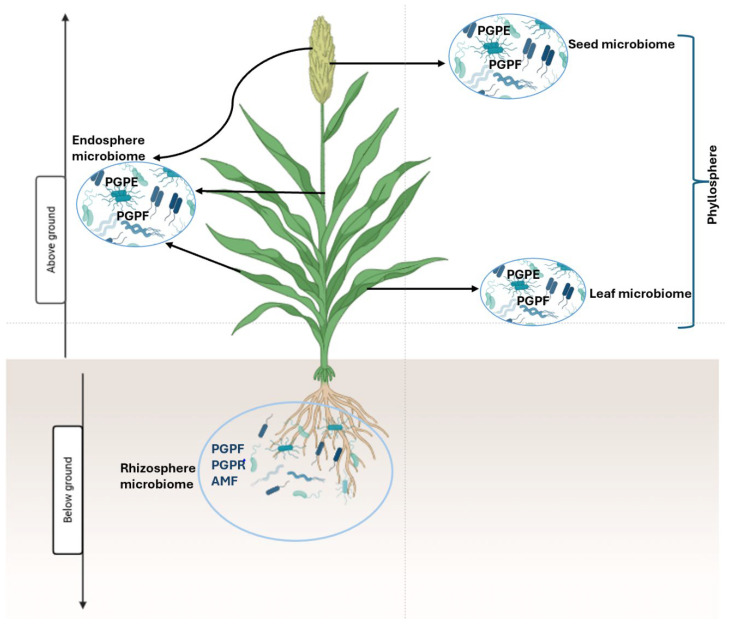
The Sorghum microbiome communities associated with the rhizosphere, endosphere, and phyllosphere.

**Table 1 plants-14-03242-t001:** Beneficial Microbes in Sorghum and Mechanism of Enhancing Growth and Productivity.

Microbe Type	Example in Sorghum	Key Mechanism of Action in Promoting Plant Growth	Reference
Plant growth-promoting rhizobacteria (PGPR)	*Pseudomonas* spp., *Rhizobium* spp., *Azospirillum* spp., *Enterobacter* spp., *Bacillus* spp., *Paenibacillus* spp., *Amycolatopsis* spp., and *Streptomyces* spp.	They promote plant growth and productivity through phosphate (P) solubilisation, nitrogen (N) fixation, and siderophore production, inducing the expression of phytohormones and enhancing the synthesis of antioxidant enzymes.	[26,27,28]
Arbuscular mycorrhizal fungi (AMF)	*Gigaspora* spp., *Scutellospora* spp., *Glomus* spp., *Sclerocystis* spp., *Entrophospora* spp., and *Acaulospora* spp.	Forms an extensive hyphal network and symbiotic relationship with sorghum roots to enhance water and nutrient uptake.	[29,30]
Plant growth-promoting fungi (PGPF)	*Trichoderma* spp., *Penicillium* spp., and *Aspergillus* spp.	Contributes to plant growth through diverse mechanisms, including the synthesis of secondary metabolites and plant-growth-promoting substances.	[31,32]
Plant growth-promoting endophytes (PGPE)	*Bacillus* spp., *Pseudomonas* spp., *Burkholderia* spp., *Micrococcus* spp., *Stenotrophomonas* spp., and *Pantoea* spp.	They colonise internal plant tissues and synthesise bioactive compounds, as well as growth-promoting hormones. They induce the expression of stress response genes that promote plant growth and stress resilience.	[26]

**Table 2 plants-14-03242-t002:** Summary of The Effects of Sorghum-Associated Microbiomes on Crop Productivity, Proposed Mechanisms, Key Outcomes and Limitations.

Microbial Group	Representative Taxa	Study Type	Source of Isolate	Crop Treated	Target Stress	Mechanism of Action	Key Outcomes	Key Limitations	Ref
PGPE	*Enterobacter* spp., *Klebsiella* spp., *Pantoea* spp.	Greenhouse (51 days)	Sorghum roots	Sorghum	None	Increased auxin production; ACC deaminase activity	Increased shoot biomass and nitrogen concentration	Controlled environment; short duration (51 days); quantitative increase over control not specified.	[48]
PGPE	*Rhizobium* spp., *Pantoea* spp., *Enterobacter* spp., *Bacillus* spp.	Greenhouse (3 months)	Sorghum root, stem and seeds	Sorghum	None	IAA production; nitrogen fixation	10–35% increase in shoot, root, and stem biomass	Short-term greenhouse trial only	[47]
PGPR	*Acinetobacter pittii*	Greenhouse (1 month)	Sorghum rhizosphere soil	Sorghum	None	IAA and siderophore production; P and potassium (K) solubilisation	Improved morphological, physiological, and biochemical traits	Controlled environment; short duration (1 month); quantitative increase over control not specified	[51]
PGPE	*Bacillus* spp., *Paenibacillus intermedius*, *A. pittii*	*In vitro* (7 days)	Sorghum root	Sorghum	Drought (PEG-induced)	Osmolyte (Proline accumulation); Exopolysaccharides production; IAA and GA production	>20% increase in germination; >30% increase in biomass	Drought stress was simulated using PEG (6000) for 7 days; Not validated *in planta*.	[56]
PGPR	*Bacillus* spp.	Greenhouse (47 days)	Field-grown sorghum	Sorghum	Drought	Not specified	Increased nitrogen accumulation; improved photosynthesis & transpiration	Controlled environment; short duration (47 days); precise mechanism not elucidated.	[57]
PGPR	*Bacillus* spp.	Greenhouse	Sorghum rhizosphere soil	Sorghum	Moisture stress	Not specified	Increased shoot length, root biomass, chlorophyll, proline, and sugar content.	Greenhouse study only; no field validation.	[54]
PGPR	*Pseudomonas* spp., *Klebsiella* spp., *Bacillus* spp., *Enterobacter* spp.	*In vitro* (3 days)	Sorghum farm soil	Sorghum	Weed (*Striga* spp.)	Hydrogen cyanide and IAA production	Reduction of Striga seed germination to 0% in vitro.	*In vitro* assay only (3 days); not tested in soil or field conditions.	[58]
PGPF + PDPE	*Trichoderma* spp., *Pseudomonas* spp., *Bacillus* spp.	Greenhouse + field	Sorghum rhizosphere soil	Sorghum	Bacterial wilt (*Dickeya dadantii*)	Callose/lignin deposition; pathogen inhibition	>30% disease reduction; 19–36% increase shoot length; 33–78% root biomass; ~30% yield increase	Field validation limited to 2 seasons; Field consistency unknown (2-season trial).	[53]
PGPE	*Trichoderma asperellum*, *Epicoccum nigrum*, *Alternaria longipes*	Greenhouse + field	Sorghum root, stem, seed	Sorghum	Fungal pathogens (*Fuariun thapsinum*, *Epicoccum sorghinum*, *Alternaria alternata*, *Curvularia lunata*)	Antifungal activity; competition; host resistance induction; siderophore production; P solubilisation.	>90% increase in germination and yield	Field validation limited (2 seasons, 1 location)	[31]
PGPE	*Trichoderma* spp.	Greenhouse + field	Sorghum farm soil	Sorghum	Fungi (*Colletotrichum graminicola*)	Antioxidant defence induction; root lignification	>50% disease reduction; >20% yield increase	Field validation limited (2 seasons, 1 location)	[49]
PGPE	*Azospirillum*, *Acetobacter*, *Trichoderma*	Field (90 days)	Sorghum rhizosphere soil	Sorghum	None	Enhanced nutrient acquisition via P solubilisation	~50% improved germination; increased grain yield	Short-term field trial(90 days); single location	[59]
PGPR	*Bacillus* spp.	Greenhouse (4 weeks)	Sorghum rhizosphere soil	Sorghum	Fungi (*Fusariun oxysporum*)	Direct competition with pathogenic fungi for root colonisation	70–100% disease reduction;	Greenhouse only; short duration (4 weeks).	[60]
PGPR	*Pseudomonas geniculat*, *Rhizobium* *pusense*, and *Bacillus* spp.	Greenhouse (8 weeks)	Rhizosphere and non-rhizosphere sorghum soil	Sorghum	Not applicable	Phytohormone production; phosphate solubilisation	>20% increase in root and shoot; 50–160% increase in photosynthetic pigment.	Greenhouse only; short duration (8 weeks)	[61]
PGPR	Not specified	Greenhouse (30 days)	Sorghum rhizosphere	Sorghum	Not applicable	Phytohormone production; phosphate solubilisation; ACC deaminase production	100% increase in germination rate.	Greenhouse only; short duration (30 days); the names of the isolates was not specified.	[52]

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
