# Peer review of "Harnessing the Sorghum Microbiome for Enhancing Crop Productivity and Food Security Towards Sustainable Agriculture in Smallholder Farming"

_plants, 2025, doi:10.3390/plants14213242_

Round 1
Reviewer 1 Report
Comments and Suggestions for Authors
The paper title " Harnessing the sorghum microbiome for enhancing crop productivity and food security towards sustainable agriculture in smallholder farming" by Omolola Aina and Lara Donaldson submitted to Plants reviewed the critical role of the sorghum microbiome in promoting sustainable crop production, particularly within smallholder farming systems. Certainly, there is strong evidence to suggest that this review will achieve a high citation rate in the future due to its potential influence on practical applications. However, some minor issues need to be improved at present.
Table 1 & L190-202: Endophytes -> plant growth-promoting endophytes (PGPE)
It is important to change the term. The term "Endophytes" denotes an ecological classification based on a microbe's habitat (residing within plant tissues), whereas "Plant-growth promoting" represents a functional classification based on a microbial activity. A significant proportion of PGPF strains functionally belong to the ecological group of endophytes.
Figure 1: Why are PGPRs absent in the rhizosphere? Furthermore, why are endophytes included within the concept of the rhizosphere microbiome?
Would it be better to relocate Section 3 after Section 5, as this arrangement would enhance the overall logical flow of the manuscript.
There are many parts (specialized words & species) in the manuscript that should be italicized but are not. Please review and correct them throughout the text.
Author Response
Reviewer 1
Comments and Suggestions for Authors
The paper title “Harnessing the sorghum microbiome for enhancing crop productivity and food security towards sustainable agriculture in smallholder farming” by Omolola Aina and Lara Donaldson submitted to Plants reviewed the critical role of the sorghum microbiome in promoting sustainable crop production, particularly within smallholder farming systems. Certainly, there is strong evidence to suggest that this review will achieve a high citation rate in the future due to its potential influence on practical applications. However, some minor issues need to be improved at present.
Comment 1:
Table 1 & L190-202: Endophytes -> plant growth-promoting endophytes (PGPE)
It is important to change the term. The term “Endophytes” denotes an ecological classification based on a microbe’s habitat (residing within plant tissues), whereas “Plant-growth promoting” represents a functional classification based on a microbial activity. A significant proportion of PGPF strains functionally belong to the ecological group of endophytes.
Response:
We thank the reviewer for this important clarification. In line with the suggestion, the term “endophytes” has been replaced with “plant growth-promoting endophytes (PGPE)” throughout the manuscript to reflect the functional classification better.
Comment 2:
Figure 1: Why are PGPRs absent in the rhizosphere? Furthermore, why are endophytes included within the concept of the rhizosphere microbiome?
Response:
We thank the reviewer for this valuable observation. Figure 1 has been revised to correctly depict the rhizosphere microbiome. Endophytes have been removed from the rhizosphere section, and PGPRs have been appropriately represented.
Comment 3:
Would it be better to relocate Section 3 after Section 5, as this arrangement would enhance the overall logical flow of the manuscript.
Response:
We appreciate this constructive suggestion. Section 3 has now been moved to follow Section 5 to improve the overall logical flow of the manuscript.
Comment 4:
There are many parts (specialised words & species) in the manuscript that should be italicised but are not. Please review and correct them throughout the text.
Response:
We thank the reviewer for bringing this issue to our attention. All scientific names, species names, and specialised terms have now been carefully checked and italicised throughout the manuscript.
Reviewer 2 Report
Comments and Suggestions for Authors
The application of microbiome in crop production is an environmental friendly and sustainable way. The authors reviewed the work on sorghum microbiome, and point out the challenges in this area. It's a good topic. However, the manuscript now have some defects:
Comments on the Quality of English LanguageThe application of microbiome in crop production is an environment friendly and sustainable way. The authors reviewed the work on sorghum microbiome, and its benefits in small holder farming system, and also point out the challenges in this area. It's a good topic. However, there are some defects of the manuscript (MS) now:
- From subtitle, it’s hard to find out which paragraph is closely related to the title “Harnessing the Sorghum Microbiome for Enhancing Crop Productivity and Food Security towards Sustainable Agriculture in smallholder farming”. Maybe some field practices need to be given.
- The paragraph such as “4. Isolation and Characterisation of the Sorghum Microbiome” are too primary, too basic, less of new information.
- The MS prepare is insufficient, most of scientific name of organisms are not italic. For example, line 251, 311-312, 343, etc.
Author Response
Reviewer 2
Comments on the Quality of English Language
The application of microbiome in crop production is an environment friendly and sustainable way. The authors reviewed the work on sorghum microbiome, and its benefits in small holder farming system, and also point out the challenges in this area. It’s a good topic. However, there are some defects of the manuscript (MS) now:
Comment 1:
From subtitle, it’s hard to find out which paragraph is closely related to the title “Harnessing the Sorghum Microbiome for Enhancing Crop Productivity and Food Security towards Sustainable Agriculture in smallholder farming”. Maybe some field practices need to be given.
Response:
- We thank the reviewer for this important observation. We agree that strengthening the connection between scientific evidence and its practical field application is crucial for a manuscript with this title. In response, we have substantially revised Section 3 to ensure that the narrative explicitly links microbiome research to the context of smallholder farming. Key changes include:
- We have changed the section title was changed from “Benefits of the Sorghum Microbiome in Smallholder Farming Systems” to “Prospects of the Sorghum Microbiome for Improved Crop Productivity in Smallholder Farming” to better reflect a critical, forward-looking analysis of potential rather than a simple list of benefits.
- We rewrote the section from a descriptive literature review into a critical synthesis that evaluates experimental evidence against field applicability.
- We integrated more greenhouse- and field-based studies of sorghum-associated microbes (highlighted in the revised Table 2) to illustrate practical applications.
- We explicitly highlighted the context-dependency of these studies and the gap between greenhouse promise and field-level consistency, framing these as key implementation challenges.
- We incorporated examples of microbial inoculant performance under real-world and smallholder-relevant conditions (e.g. multi-season field trials, effects on yield and nutrient uptake).
Comment 2:
The paragraph such as “4. Isolation and Characterisation of the Sorghum Microbiome” are too primary, too basic, less of new information.
Response:
We thank the reviewer for this valuable comment. We agree that the initial version of the “Isolation and Characterisation of the Sorghum Microbiome” section was overly descriptive and provided mainly basic information. In the revised manuscript, we have substantially restructured this section to go beyond standard isolation techniques and to highlight recent methodological advances that are particularly relevant for sorghum. Specifically, we have:
- Reduced the generic description of routine culture-based isolation methods and added examples specific to sorghum-associated microbes with plant-growth-promoting or biocontrol traits.
- Introduced culture-independent approaches such as molecular fingerprinting (DGGE, T-RFLP) and high-throughput sequencing, showing how they complement conventional isolation in capturing unculturable and syntrophy-dependent taxa.
- Added recent omics-based studies (metagenomics, proteomics, metabolomics) that illustrate how functional genes, metabolic pathways and bioactive compounds are now being discovered directly from the sorghum microbiome.
- Provided concrete case studies (e.g. identification of nifH-harbouring bacteria; recovery of nitrogen-fixing strains from field-grown sorghum) to demonstrate the application of these methods in real research contexts.
Comment 3:
The MS prepare is insufficient, most of scientific name of organisms are not italic. For example, line 251, 311-312, 343, etc.
Response:
We thank the reviewer for drawing our attention to this issue. All scientific names, species names, and specialised terms have now been carefully checked and italicised throughout the manuscript.
Reviewer 3 Report
Comments and Suggestions for Authors
Review report
The manuscript addresses a timely and important topic: leveraging the sorghum microbiome to improve productivity and resilience in smallholder systems. The narrative is comprehensive in scope (composition, recruitment, isolation/characterization, agronomic benefits, challenges, and policy/implementation pathways) and is well aligned with sustainability goals. However, to meet the standards of a critical review, the paper needs (i) transparent review methodology, (ii) a more balanced appraisal of evidence (including limitations and negative results), (iii) tighter and more precise claims, and (iv) significant improvements to structure, figures/tables, and referencing.
- Add a Methods for the review (databases searched, time window, keywords, inclusion/exclusion criteria, language limits). A PRISMA-style flow diagram or at least a textual description will improve reproducibility and reduce selection bias.
- Add evidence tables summarizing key trials (location, sorghum genotype, inoculant identity/CFU, design, environment, response variables, effect sizes with variance, duration). Separate greenhouse vs field outcomes.
- Provide practical targets (carrier types, moisture range, pH, typical label claims, viability thresholds, CFU load per seed or per hectare, accelerated-aging tests, expected shelf life at ambient temperatures). Add indicative costs (per hectare), logistics (cold chain vs ambient), and return-on-investment scenarios; contrast with subsidized mineral inputs.
- P1L10 - Sorghum (Sorghum bicolor L. Moench) – correct Sorghum bicolor Moench
- Section 3:
- The section mostly summarizes positive findings from the literature without weighing the strength, scale, or limitations of those studies (e.g., greenhouse vs field, small sample size, lack of long-term trials). There is no mention of variability in outcomes (cases where inoculants failed, context-dependency, or farmer-level constraints).
- No effort to quantify effects (ranges of yield increase, stress tolerance improvements) or compare across microbial groups (PGPR vs AMF vs Trichoderma).
- Mechanisms are described, but the section does not highlight which are most consistently supported in sorghum or which remain speculative.
- Lack of information on economic feasibility of inoculants, practical application methods, or availability in real smallholder contexts.
- Almost no coverage of ecological risks (e.g., displacement of native strains, pathogen contamination, inconsistency in formulations).
- Subsections (“Addressing Financial Constraints…”, “Addressing Poor Soil Fertility…”, “Addressing Climate Variability…”) overlap conceptually and repeat the same logic: inoculants help with stress → better growth → higher yields. This makes the section long, redundant, and less impactful.
- The narrative lacks a clear hierarchy—findings, mechanisms, and broader implications are mixed together.
- Some references are used only to support generic statements (e.g., low soil fertility, drought impacts) rather than sorghum-specific evidence. In some cases, the mechanistic detail is excessive (e.g., enzyme names, biochemical pathways) without connecting back to practical significance. This risks overwhelming readers without adding synthesis.
Author Response
Reviewer 3
Comments and Suggestions for Authors
The manuscript addresses a timely and important topic: leveraging the sorghum microbiome to improve productivity and resilience in smallholder systems. The narrative is comprehensive in scope (composition, recruitment, isolation/characterisation, agronomic benefits, challenges, and policy/implementation pathways) and is well aligned with sustainability goals. However, to meet the standards of a critical review, the paper needs (i) transparent review methodology, (ii) a more balanced appraisal of evidence (including limitations and negative results), (iii) tighter and more precise claims, and (iv) significant improvements to structure, figures/tables, and referencing.
Comment 1
Add a Methods for the review (databases searched, time window, keywords, inclusion/exclusion criteria, language limits). A PRISMA-style flow diagram or at least a textual description will improve reproducibility and reduce selection bias.
Response:
We thank the reviewer for this valuable suggestion. In response, we have added a dedicated Methods section that clearly describes our literature search strategy, including the databases searched. We also provide a textual description of the screening process, specifying the number of articles initially retrieved, excluded, and retained for full-text review.
Comment 2:
Add evidence tables summarising key trials (location, sorghum genotype, inoculant identity/CFU, design, environment, response variables, effect sizes with variance, duration). Separate greenhouse vs field outcomes.
Response:
We thank the reviewer for the valuable suggestion. We have included a table which summarises recent studies of the effects of sorghum-associated microbiomes on crop productivity and incorporated all the suggestions. The inoculant identity/CFU was not included as this was not provided in most of the studies.
Comment 3:
Provide practical targets (carrier types, moisture range, pH, typical label claims, viability thresholds, CFU load per seed or per hectare, accelerated-aging tests, expected shelf life at ambient temperatures). Add indicative costs (per hectare), logistics (cold chain vs ambient), and return-on-investment scenarios; contrast with subsidised mineral inputs.
Response:
We thank the reviewer for this highly constructive suggestion. We fully agree that providing concrete technical specifications and economic considerations is essential to demonstrate the practical feasibility of microbial inoculants in smallholder systems. In response, we have substantially expanded the subsection on “Formulation and Application Strategies for Sorghum Microbial Inoculants,” with two new, dedicated paragraphs that provide the following details;
- We included formulation benchmarks, including typical viability thresholds, pH range, target shelf life, and CFU loading targets, both at the seed level and per hectare.
- We analysed a direct comparison of indicative costs per hectare to highlight the affordability of microbial inoculants relative to mineral fertilisers.
Comment 4:
P1L10 - Sorghum (Sorghum bicolor L. Moench) – correct Sorghum bicolor Moench
Response:
We thank the reviewer for drawing our attention to this issue. This has been corrected throughout the manuscript.
Comment 5:
The section mostly summarises positive findings from the literature without weighing the strength, scale, or limitations of those studies (e.g., greenhouse vs field, small sample size, lack of long-term trials). There is no mention of variability in outcomes (cases where inoculants failed, context-dependency, or farmer-level constraints).
Response:
We thank the reviewer for this crucial feedback. We agree that a critical evaluation of the evidence is essential for a balanced perspective on the true prospects of sorghum microbiome applications. In direct response to this comment, we have completely reframed the section from a descriptive summary into a critical synthesis. The following major revisions were made to address the specific points raised:
- We have integrated a critical assessment of the strength of evidence throughout the narrative. The section now explicitly differentiates between greenhouse/pot trials and field-validated results.
- We introduced Table 2, which includes a dedicated “Key Limitations” column. This column systematically categorises the constraints of each cited study and reported the efficacy of the studies.
- We eliminated overlapping subsections and structured the section to first present the synthesised evidence, then critically evaluate the mechanisms, and finally conclude with the translational gaps to avoid the previously noted redundancy,
Comment 6:
No effort to quantify effects (ranges of yield increase, stress tolerance improvements) or compare across microbial groups (PGPR vs AMF vs Trichoderma).
Response:
We thank the reviewer for the valuable suggestion. We included a section in the newly introduced table to quantify the observed effects, including yield increase and the level of stress tolerance improvement.
Comment 7:
Mechanisms are described, but the section does not highlight which are most consistently supported in sorghum or which remain speculative.
Response:
We thank the reviewer for this insightful comment. We agree that evaluating the strength of evidence for different mechanisms is crucial for prioritising research and application efforts. We have added a new subsection that highlights the mechanisms frequently reported in sorghum. We also included a section in Table 2 which summarises the methods reported in each study.
Comment 8:
Lack of information on economic feasibility of inoculants, practical application methods, or availability in real smallholder contexts.
Response:
We thank the reviewer for this valuable suggestion. We have structured the manuscript so that Section 3 focuses on the agronomic prospects based on experimental evidence, while the critical issues of economic feasibility, practical application, and ecological risks are addressed in detail in dedicated later sections (Sections 5 and 6). We have now added a concluding sentence in Section 3 to explicitly signpost this structure for the reader
Comment 9:
Almost no coverage of ecological risks (e.g., displacement of native strains, pathogen contamination, inconsistency in formulations).
Response:
We thank the reviewer for raising this critical point regarding the ecological considerations of applying microbial inoculants. We agree that a discussion of potential risks is essential for a balanced and responsible review. In our manuscript structure, we dedicated the section in question (Section 3) to focus on synthesising the potential of the sorghum-microbiome from scientific literature. We have significantly strengthened Section 6.0 (“Challenges and Future Directions”) with a dedicated paragraph that explicitly discusses the potential ecological impacts, such as introducing microbial strains that may compete with or replace native microbial strains.
Comment 10:
Subsections (“Addressing Financial Constraints…”, “Addressing Poor Soil Fertility…”, “Addressing Climate Variability…”) overlap conceptually and repeat the same logic: inoculants help with stress → better growth → higher yields. This makes the section long, redundant, and less impactful.
Response:
We thank the reviewer for this excellent observation. We agree that the previous subsection structure created an unnecessarily repetitive narrative. The section has been restructured and is now titled “Prospects of the Sorghum Microbiome for Improved Crop Productivity in Smallholder Farming.” We collapsed all subheadings into a unified section, which presents a critical synthesis of the evidence, organised thematically.
Comment 11:
The narrative lacks a clear hierarchy—findings, mechanisms, and broader implications are mixed together.
Response:
We thank the reviewer for this critical assessment of the section’s narrative flow. We agree that the previous intermingling of findings, mechanisms, and implications reduced the clarity and impact of the argument. To address this, we have fundamentally restructured the section to establish a clear and logical hierarchy. The revised narrative now follows a sequential and systematic progression, starting with a quantitative overview of the key findings from the literature, analysis of mechanisms, and concluding with he limitations
Comment 12:
Some references are used only to support generic statements (e.g., low soil fertility, drought impacts) rather than sorghum-specific evidence. In some cases, the mechanistic detail is excessive (e.g., enzyme names, biochemical pathways) without connecting back to practical significance. This risks overwhelming readers without adding synthesis.
Response:
We thank the reviewer for these precise and valuable suggestions. We have carefully revised the manuscript to address both concerns. We have replaced generic citations with sorghum-specific studies, where possible. We have streamlined the discussion of mechanisms, removing overly detailed biochemical lists where they did not contribute to the main argument. The focus is now on the functional outcome of these mechanisms for the plant.
Round 2
Reviewer 2 Report
Comments and Suggestions for Authors
The MS was improved greatly and better than previous version, while, there are still some problems:
- The authors added a table (should be table 2), however, there is no table legend. I
- n scientific paper, table usually presented in three-line table. However, Table 1 and Table 2 are not three-line table, and they were presented in different type.
- The other mistakes: such as line 258 “Trichoderma spp.” Should be “Trichoderma spp.”, “spp.” Should not be italicized, Etc.
Author Response
Thank you very much to the reviewer for picking up these final errors.
1) We have now included a title for the new Table 2.
2) We have reformatted the tables so that they now appear in the standard 3 line format.
3) We have corrected all occurences of spp. so that it is no longer italicized.
Thank you again for thoroughly checking the manuscript so that we could make these improvements.
Reviewer 3 Report
Comments and Suggestions for Authors
The authors have addressed all of my questions. I have no further comments. I recommend the manuscript for publication. Congratulations to the authors.
Author Response
Thank you very much to the reviewer for advising the changes which have greatly improved the manuscript. We are deeply grateful for your time, insight and advice and we are very pleased to have met your approval.